# Alcohol use during pregnancy in Rakai, Uganda

**Adriane Wynn**[1]*, **Dorean Nabukalu**[2], **Tom Lutalo**[2], **Maria Wawer**[2,3], **Larry W. Chang**[2,3,4], **Susan M. Kiene**[5], **David M. Serwadda**[6], **Nelson Sewankambo**[7], **Fred Nalugoda**[2], **Godfrey Kigozi**[2], **Jennifer A. Wagman**[8]

1 Division of Infectious Diseases & Global Public Health, Department of Medicine, University of California, San Diego, La Jolla, CA, United States of America, 2 Rakai Health Sciences Program/Uganda Virus Research Institute, Kalisizo, Uganda, 3 Department of Epidemiology, Johns Hopkins Bloomberg School of Public Health, Baltimore, MD, United States of America, 4 Department of Medicine, John Hopkins School of Medicine, Baltimore, MD, United States of America, 5 Division of Epidemiology and Biostatistics, School of Public Health, San Diego State University, San Diego, CA, United States of America, 6 Makerere University School of Public Health, Kampala, Uganda, 7 Makerere University School of Medicine, Kampala, Uganda, 8 Department of Community Health Sciences, UCLA Fielding School of Public Health, Los Angeles, CA, United States of America

* awynn@health.ucsd.edu

## Abstract

### Introduction

Antenatal alcohol use is linked to adverse maternal and neonatal outcomes. Uganda has one of the highest rates of alcohol use in sub-Saharan Africa, but the prevalence of antenatal alcohol use has not been reported in the Rakai region.

### Methods

We used cross-sectional data from pregnant women in the Rakai Community Cohort Study between March 2017 and September 2018. Using bivariate and multivariable analyses, we assessed associations between self-reported antenatal alcohol use and sociodemographic characteristics, intimate partner violence (IPV), and HIV status.

### Results

Among 960 pregnant women, the median age was 26 years, 35% experienced IPV in the past 12 months, 13% were living with HIV, and 33% reported alcohol use during their current pregnancy. After adjusting for marital status, education, smoking, and HIV status; Catholic religion (AOR: 3.54; 95% CI: 1.89–6.64; compared to other), bar/restaurant work (AOR: 2.40; 95% CI: 1.17–4.92; compared to agriculture), >one sex partner in past year (AOR: 1.92; 95% CI: 1.17–3.16), a partner that drank before sex in past year (AOR: 2.01; 95% CI: 1.48–2.74), and past year IPV (AOR: 1.55; 95% CI: 1.14–2.11) were associated with antenatal alcohol use.

### Conclusion

We found that alcohol use during pregnancy was common and associated with religion, occupation, higher numbers of past year sex partners, having a partner who drank before

**Data Availability Statement:** Data can be found online through the Rakai Community Cohort Study (RCCS) dashboard: https://www.rhsp.org/research/rccs/explore-rccs-data.

**Funding:** AW, K01 AA027733, NIAAA AW, T32 DA023356, NIDA The funders had no role in study design, data collection and analysis, decision to publish, or preparation of the manuscript.

**Competing interests:** The authors have declared that no competing interests exist.

sex in the past 12 months, and IPV experience. More research is needed to understand the quantity, frequency, and timing of antenatal alcohol use; and potential impacts on neonates; and to identify services that are acceptable and effective among pregnant women.

## Introduction

Alcohol use is an important cause of morbidity and mortality worldwide [1], and sub-Saharan Africa (SSA) bears the largest alcohol-attributable burden of disease and injury [2]. Alcohol use during pregnancy is common in many countries with one systematic review and meta-analysis estimating the worldwide prevalence to be 9.8% (8.9–11.1%) [3]. Antenatal exposure to alcohol is a global public health concern associated with adverse health consequences for women and neonates, including stillbirth, low birthweight, preterm birth, and fetal alcohol spectrum disorders (FASD) [4,5].

Uganda has one of the highest rates of alcohol use in SSA [2]. The World Health Organization (WHO) estimated that, in 2016, alcohol use per capita was 9.5 liters of pure alcohol (16.1 among men and 3.0 among women) [2]. Although 63.7% of the population (aged 15 years and older) reported that they abstained from alcohol use in the past year, among those who drank, 56.9% reported heavy episodic drinking (68.8% among men and 32.6% among women), and 7.1% met criteria for alcohol use disorders (12.4% among men and 1.9% among women) [2]. Studies of antenatal alcohol exposure in Uganda are limited. A 2010 study found that among a sample of pregnant women seeking antenatal care at the national referral hospital in Kampala (Uganda's capital city), 25% reported alcohol use after learning they were pregnant [6]. Another study that combined self-reported alcohol use and a blood biomarker found that 20% of pregnant women living with HIV in southwestern Uganda met the criteria for hazardous alcohol use [7].

Previous research in Rakai, Uganda, the setting for this research, found a high prevalence of alcohol use, including 39% among all women surveyed [8] and 42% among young women [9]. However, prevalence and correlates of alcohol use among pregnant women have not been assessed in Rakai. Understanding the patterns and drivers of alcohol use during pregnancy is important for developing interventions to reduce antenatal alcohol consumption, FASD, and interrelated problems such as HIV infection and intimate partner violence (IPV). This paper utilizes cross-sectional data collected in 2017–2018 to estimate the prevalence and correlates of antenatal alcohol use among pregnant women in Rakai, Uganda.

## Methods

### Study population

This cross-sectional study was conducted among pregnant women in the mostly rural Rakai region of southwestern Uganda. The data for this study were derived from the Rakai Community Cohort Study (RCCS) conducted between March 2017 and September 2018. RCCS is an open, prospective HIV surveillance cohort which takes place in Rakai, Uganda and has been described in detail previously [10]. Since 1994, RCCS has enrolled approximately 15,000 consenting participants aged 15–49 years every 12–20 months. Prior to each RCCS surveillance survey, a household census is conducted to identify all community members eligible for enrollment. Thereafter, all present community residents who are within the eligible age range and have provided informed consent are surveyed as part of RCCS. At each survey visit,

participants are interviewed regarding sociodemographic and behavioral information and voluntary venous blood is obtained for HIV testing. Participants who provide written informed consent are interviewed in central community locations (referred to as "hubs"). Those not captured at the hubs are approached at their household or place of work to request their participation. Up to two return visits are made in an attempt to enroll eligible participants. Interviews are administered by same sex interviewers who use a structured questionnaire with questions on sexual behaviors, sexual partners, health status and service utilization, and reproductive health. Free HIV post-test education and services are offered to participants who choose to give blood, including results to all consenting individuals and couples.

## Eligibility criteria

Data were eligible for inclusion in our analytic sample if the participant was a woman who self-reported as pregnant through an affirmative response to the survey question, "Are you pregnant now?" and who responded to the question about alcohol use during pregnancy. Although pregnancy status was confirmed by human chorionic gonadotropin (HCG) urine test for women uncertain about their pregnancy status, alcohol use during pregnancy was only assessed among those who self-reported as pregnant. Data were excluded from: (1) Women who self-reported as pregnant, but did not answer the alcohol use during pregnancy question (n = 1); and (2) Women who did not self-report as pregnant, but were later found to be pregnant via HCG urine test (n = 130). Among women who self-reported as pregnant, 213 who were not visibly pregnant were also given an HCG test, and 21 tested negative. Our sample includes all women who self-reported as pregnant and thus includes the 21 who later tested negative. We compared sociodemographic characteristics and alcohol responses between those who tested negative and those tested positive for pregnancy and found no differences.

## Measures

We assessed socio-demographic information, including age, religion, education level, occupation, and marital status. Rakai communities are also categorized as agrarian, trading or fishing. Agrarian communities are often a great distance from main roads and the most commonly reported primary occupation is agriculture and/or production and maintenance of crops and farmland. Trading communities are often close to main roads, have high mobility, and the most commonly reported primary occupation is buying and selling of goods and/or services. Fishing communities are residential areas with landing sites on Lake Victoria where the primary occupation relates to harvesting or processing of fishery resources [11]. Education level was measured using a dichotomous variable where those with zero to the seven years of primary school were defined as "full primary school or less" and those with more than seven years were defined as "secondary or higher." A household social economic status (SES) index was compiled based on dwelling structure (e.g. grass thatch roofs and mud floors were categorized as low SES) using methods developed for use in Rakai and previously described [12]. We also measured past year cigarette smoking (yes/no). HIV testing was performed using a validated three rapid test algorithm. Women living with HIV responded to the questions: "Have you ever been to a clinic to receive care for HIV?" and "Are you currently taking this medication?" (i.e., ARVs). Sexual behaviors included number of past year sex partners (none, 1, ≥2). Participants reported on whether alcohol was used by her partner(s) prior to sex in the past 12 months. IPV victimization was assessed by reports of emotional, physical, or sexual abuse by an intimate partner within the past 12 months using an adapted version of the Conflict Tactics Scale [13]. Emotional IPV was measured by asking if a partner verbally abused or shouted at the participant. Physical IPV was measured by asking if a partner perpetrated any of the

following behaviors against the participant: pushing, pulling, grabbing, kicking, slapping, punching, burning, strangling, and/or attacking with a weapon. Sexual IPV was measured by asking if a partner perpetrated any of the following sexually abusive behaviors against the participant: physically forcing her to have sex or forcing her to perform sexual activities when she did not want to.

The main alcohol use variable examined in this analysis was alcohol use during pregnancy, defined as 'Have you been drinking alcohol during this pregnancy?' (yes/no). Participants were also asked about their past year alcohol use, defined as 'Have you drunk any alcohol in the past year, for instance, beer, wine, waragi or other spirits, or home-made beer?' (yes/no). Participants who reported affirmatively to past year alcohol use were subsequently asked the following questions:

*To assess alcohol-related aggression and violence*: 1) In the past year, when you drank alcohol, did you get angry, get violent or get into a fight? (yes/no for each item).

*To assess loss of control related to drinking*: "How often during the last year have you felt you should cut down on your drinking or stop altogether?" (Never, occasionally, sometimes, often; which was recoded to Never and occasionally or more)

*To assess alcohol's impact on daily responsibilities and activities*: "In the past year, have you ever taken alcohol while you were at work?" (yes/no)

*To assess harmful alcohol use, we used three items from the Alcohol Use Disorders Identification Test (AUDIT) [14]*:

- **Guilt after drinking**: In the past year, when you drank alcohol, have you ever have felt ashamed of something that you did while drinking? (yes/no)

- **Blackouts:** In the past year, when you drank alcohol, did you ever forget some of the things you did or that happened while you were drinking? (yes/no)

- **Others concerned about drinking:** Has a relative or friend, doctor or other health worker been concerned about your drinking or suggested that you cut down? (yes/no)

  *To assess alcohol dependence*:

- Using criteria from the International Classification of Diseases (WHO, 1993): "In the past year, when you drank alcohol, did you ever experience an unsteady gait? Fall over? Have difficulty speaking? Have shaking hands the next morning?" (yes/no for each item)

- Using one item from the AUDIT scale [14]: "How often during the last year have you failed to do something that you wanted or needed to do because of your drinking?" (Never, occasionally, sometimes, often; which was recoded to Never and occasionally or more).

## Analysis

We estimated the prevalence of alcohol use during pregnancy, including a confidence interval based on a binomial distribution. Correlates of alcohol use during pregnancy were assessed using Fisher's exact or Chi-square tests for categorical variables and Wilcoxon rank-sum or Student's t-test for continuous variables. We also fit logistic regression models to calculate the unadjusted odds ratios (ORs) and 95% confidence intervals (CIs) to estimate associations between alcohol use and key sociodemographic, sexual behavior variables (e.g. number of partners), and HIV status. Next, we fit an adjusted logistic regression model, which included covariates that were significantly associated with alcohol use during pregnancy at the P<0.05 level in bivariate analyses and findings from prior research in Uganda demonstrating a relationship between alcohol use, HIV and IPV [15]. We ran collinearity diagnostics using a

Variance Inflation Factor (VIF) of $\geq 5$ or a tolerance of $\leq 0.1$ to signal the presence of multi-collinearity. In our models, multicollinearity was ruled out because tolerance measures were above 0.85 and all VIFs did not exceed two.

We also stratified the alcohol-related consequence variables by alcohol use during pregnancy and experience of past year IPV and conducted bivariate comparisons using Fisher's exact or Chi-square tests for categorical variables and Wilcoxon rank-sum or Student's t-test for continuous variables.

### Ethics

All study instruments and protocols were reviewed and approved by the Western IRB, the Uganda Virus Research Institute's Research and Ethics Committee (UVRI-REC) and the Uganda National Council of Science and Technology (UNCST). The plan for analysis and publication of these data was reviewed and approved by the University of California (UC) San Diego Human Research Protections Program (HRPP), the UC Los Angeles HRPP, and UVRI-REC. All RCCS participants provided written consent to take part in the study and all were compensated 10,000 Ugandan shillings (approximately $3 USD) for their time and transport refund.

### Results

In our sample of 960 women who reported they were pregnant at the time of interview, 33% (95% CI: 30–36%) reported that they used alcohol during their current pregnancy. Table 1 displays sociodemographic characteristics, HIV status, HIV care and ART uptake, number of sex partners, whether partners drank before sex, and IPV experiences stratified by alcohol use during pregnancy. The median age was 26 years, 81% were currently married, 63% had completed full primary school or less, and 59% were Roman Catholic. The most commonly reported occupation among women was in agriculture (41%) followed by trader or shopkeeper (24%) and housework (in own home) (16%). The largest proportion of participants (55%) were drawn from agrarian communities; 24% were from trading centers, and 21% were from fishing villages. Only 2% of women reported that they currently smoked.

The HIV prevalence was 13%. Among women living with HIV, 94% ever had HIV care and 91% reported that they were on ART at the time of interview. Regarding sex partners, 10% reported having two or more sex partners in the past 12 months, and 39% reported that partner(s) drank alcohol before sex in the past 12 months. In the past year, 35% of women reported that they experienced some form of IPV, including emotional (32%), physical (23%), and sexual (7%).

In the unadjusted models, drinking during pregnancy was associated (at the $p < 0.05$ level) with being previously married, having a lower level of education, Catholic religion, being employed at a bar or restaurant, past year smoking, having two or more sex partners in the past year, having a partner that drank before sex in the past year, living with HIV, and experiencing IPV in the past year (Table 2). In the adjusted model, Catholic religion (AOR: 3.54; 95% CI: 1.89–6.64), having a job in housework (AOR: 1.69; 95% CI: 1.09–2.61), having a job at a bar or restaurant (AOR: 2.40; 95% CI: 1.17–4.92), having two or more sex partners in the past year (AOR: 1.92; 95% CI: 1.17–3.16), having a partner that drank before sex in the past year (AOR: 2.01; 95% CI: 1.48–2.74), and past year IPV (AOR: 1.55; 95% CI: 1.14–2.11) were associated with alcohol use during pregnancy. Marital status, education level, past year smoking, and HIV status were no longer associated with alcohol use during pregnancy after adjusting for covariates.

**Table 1. Characteristics by alcohol use during pregnancy among participants in RCCS between March 2017 and September 2018.**

| | Total Sample | Have you been drinking during this pregnancy? | | |
|---|---|---|---|---|
| | (N = 960) | Yes (n = 319) (33%) | No (n = 641) (67%) | |
| | No. (col %) | No (row %) | No (row %) | P-value |
| **Age (years), Median [range]** | 26 [15–46] | 26 [16–46] | 26 [15–42] | 0.409 |
| 15–24 yrs | 425 (44) | 137 (32) | 288 (68) | |
| 25–34 yrs | 399 (42) | 133 (33) | 266 (67) | |
| 35+ yrs | 136 (14) | 49 (36) | 87 (64) | |
| **Marital status** | | | | **0.021** |
| Currently married | 786 (81) | 255 (32) | 531 (68) | |
| Previously married | 101 (11) | 45 (45) | 56 (55) | |
| Never | 73 (8) | 19 (26) | 54 (74) | |
| **Education** | | | | **0.02** |
| Primary or less | 602 (63) | 217 (36) | 385 (64) | |
| Secondary or higher | 358 (37) | 102 (28) | 256 (72) | |
| **Religion** | | | | **<0.001** |
| Other | 80 (8) | 15 (19) | 6 (81) | |
| Catholic | 568 (59) | 228 (40) | 340 (60) | |
| Protestant | 166 (17) | 47 (28) | 119 (72) | |
| Muslim | 146 (15) | 29 (20) | 117 (80) | |
| **Occupation** | | | | **0.004** |
| Agriculture | 391 (41) | 125 (32) | 266 (68) | |
| Housework or Housekeeper | 149 (16) | 59 (40) | 90 (60) | |
| Clerical/Teacher | 62 (6) | 12 (19) | 50 (81) | |
| Trader/Shop keeper | 229 (24) | 77 (34) | 152 (66) | |
| Bar or Restaurant | 41 (4) | 22 (54) | 19 (46) | |
| Other | 88 (9) | 24 (27) | 64 (73) | |
| **Household SES** | | | | 0.225 |
| High | 633 (66) | 210 (33) | 423 (67) | |
| Middle | 195 (20) | 58 (30) | 137 (70) | |
| Low | 131 (14) | 51 (39) | 80 (61) | |
| **Community type** | | | | 0.067 |
| Agrarian | 532 (55) | 177 (33) | 355 (67) | |
| Trading | 230 (24) | 65 (28) | 165 (72) | |
| Fishing | 198 (21) | 77 (39) | 121 (61) | |
| **Currently smoke** | | | | **0.002** |
| Yes | 18 (2) | 12 (67) | 6 (33) | |
| No | 942 (98) | 307 (33) | 635 (67) | |
| **HIV Status** | | | | **0.026** |
| Positive | 127 (13) | 53 (42) | 74 (58) | |
| Negative | 822 (87) | 261 (32) | 561 (68) | |
| **Ever had HIV care** (HIV positive) | | | | 0.718 |
| Yes | 119 (94) | 49 (41) | 70 (59) | |
| No | 8 (6) | 4 (50) | 4 (50) | |
| **Currently on ART** (HIV positive) | | | | 0.236 |
| Yes | 115 (91) | 46 (40) | 69 (60) | |
| No | 12 (9) | 7 (58) | 5 (42) | |
| **Number of sex partners past year** | | | | **<0.001** |
| 1 | 864 (90) | 266 (31) | 598 (69) | |

(*Continued*)

**Table 1.** (Continued)

| | Total Sample | Have you been drinking during this pregnancy? | | |
| --- | --- | --- | --- | --- |
| | (N = 960) | Yes (n = 319) (33%) | No (n = 641) (67%) | |
| | No. (col %) | No (row %) | No (row %) | P-value |
| 2+ | 95 (10) | 52 (55) | 43 (45) | |
| **Partner drank before sex past year** | | | | |
| Yes | 375 (39) | 174 (46) | 201(54) | <0.001 |
| No | 584 (61) | 145 (25) | 439 (75) | |
| **Intimate Partner Violence (IPV) past year** | | | | <0.001 |
| No IPV | 662 (65) | 174 (28) | 448 (72) | |
| Any IPV | 337 (35) | 145 (43) | 192 (57) | |
| Emotional IPV | 308 (32) | 136 (44) | 172 (56) | |
| Physical IPV | 217 (23) | 96 (44) | 121 (55) | |
| Sexual IPV | 69 (7) | 30 (43) | 39 (57) | |

Notes: Ever had HIV care and currently on ART are among those living with HIV. IPV p-value is between none and any IPV. Pregnant at time of abuse and partner drank at time of abuse are among those who experienced emotional, physical or sexual abuse.

Among pregnant women who reported drinking in the past year (n = 386, 40%), 96 (25%) reported that they did not drink during their current pregnancy and 280 (73%) reported experiencing no alcohol-related consequence (Table 3). The most common consequences of alcohol use were related to alcohol dependence and impact on daily responsibilities, including feeling they should cut down on drinking (34%), failing to do something they wanted or needed to do (13%), drinking at work (13%), and having an unsteady gait (12%). Few significant differences emerged in terms of alcohol-related consequences between past year drinkers who did and did not consume alcohol during their pregnancy. One exception is that women who reported drinking in the past year but *not* during their pregnancy were more likely to report an unsteady gait in the past year (19% compared to 10%). We also compared the reported consequences of past year alcohol use by IPV experience (among women who drank in the past year). Women who experienced IPV had significantly higher proportions of alcohol consequences compared to those who didn't experience IPV, with the exception of the variable "did you ever fall over," where only three people said yes. Among women who drank in the past year and experienced IPV, the three most commonly reported alcohol-related consequences were feeling they should cut down on their drinking (42%), failing to do something they wanted or needed (21%), and getting angry (20%).

## Discussion

### Our findings

We found that the prevalence of self-reported drinking during pregnancy in Rakai, Uganda, was over three times higher than the global average of alcohol use among pregnant women [3], higher than previous studies conducted among pregnant women in other regions of Uganda [6,7], but lower than the prevalence of alcohol use among non-pregnant women in Rakai, Uganda [8]. Correlates of alcohol use during pregnancy were Catholic religion, occupation in a restaurant or bar, higher numbers of past year sex partners, having a partner who drank before sex in the past 12 months, and experiencing IPV. Among women who reported drinking in the past year, consequences of alcohol use (e.g. feeling you should cut down drinking, failing to something you needed because of drinking, and getting angry when drinking) were

**Table 2. Unadjusted and adjusted logistic regression models of correlates of alcohol use during pregnancy among participants in RCCS between March 2017 and September 2018.**

| | Unadjusted Estimate | | | Adjusted Estimate | | |
|---|---|---|---|---|---|---|
| | Odds Ratio | 95% CI | p-value | Odds Ratio | 95% CI | p-value |
| **Marital status** | | | | | | |
| Never married | Ref | | | | | |
| Previously married | 2.28 | (1.19–4.39) | **0.01** | 1.47 | (0.72–3.02) | 0.29 |
| Currently married | 1.36 | (0.79–2.35) | 0.26 | 1.16 | (0.65–2.08) | 0.62 |
| **Education** | | | | | | |
| ≤Primary | Ref | | | | | |
| ≥Secondary | 0.71 | (0.53–0.94) | **0.02** | 0.85 | (0.61–1.18) | 0.33 |
| **Religion** | | | | | | |
| Other | Ref | | | | | |
| Catholic | 2.91 | (1.62–5.22) | **<0.001** | 3.54 | (1.89–6.64) | **<0.001** |
| Protestant | 1.71 | (0.89–3.30) | 0.12 | 1.94 | (0.96–2.80) | 0.06 |
| Muslim | 1.07 | (0.54–2.15) | 0.2 | 1.34 | (0.63–2.80) | 0.77 |
| **Occupation** | | | | | | |
| Agriculture | Ref | | | | | |
| Housework | 1.4 | (0.94–2.06) | 0.1 | 1.69 | (1.09–2.61) | **0.02** |
| Clerical/Teacher | 0.51 | (0.26–0.99) | **0.05** | 0.76 | (0.37–1.59) | 0.47 |
| Trader/Shop | 1.08 | (0.76–1.53) | 0.67 | 1.34 | (0.91–1.97) | 0.13 |
| Bar or Restaurant | 2.46 | (1.29–4.72) | **0.007** | 2.4 | (1.17–4.92) | **0.02** |
| Other | 0.8 | (0.48–1.34) | 0.39 | 0.84 | (0.48–1.48) | 0.55 |
| **Currently smoke** | | | | | | |
| No | Ref | | | | | |
| Yes | 4.14 | (1.54–11.13) | **0.005** | 2.82 | (0.98–8.10) | **0.05** |
| **HIV Infected** | | | | | | |
| Uninfected | Ref | | | | | |
| Infected | 1.54 | (1.05–2.26) | **0.03** | 1.01 | (0.66–1.56) | 0.96 |
| **Number of sex partners past year** | | | | | | |
| 1 | Ref | | | | | |
| 2+ | 2.77 | (1.80–4.25) | **<0.001** | 1.92 | (1.17–3.16) | **0.01** |
| **Partner drank before sex** | | | | | | |
| No | Ref | | | | | |
| Yes | 2.62 | (1.99–3.46) | **< .001** | 2.01 | (1.48–2.74) | **<0.001** |
| **Intimate Partner Violence (IPV)** | | | | | | |
| No IPV | Ref | | | | | |
| Any IPV | 1.94 | (1.47–2.57) | **< .001** | 1.55 | (1.14–2.11) | **0.005** |

more common among women who also experienced IPV during the past year, compared to women who drank in the past year but did *not* experience IPV.

Our findings bolster previous results demonstrating the co-occurrence of and interrelationship between alcohol use and syndemics such as IPV, depression, and HIV infection among non-pregnant women in SSA [15–17]. A recent study found that 47% of non-pregnant women attending a hospital outpatient clinic and eligible for HIV testing in rural Uganda reported having one or more of the following conditions: depression, emotional or physical IPV, and/or alcohol use [15]. This study also found evidence of synergistic effects of multiple conditions to increase HIV risk. Women experiencing two or more conditions (compared to none) reported more high-risk sex acts (AOR: 2.18; 95% CI: 1.64–2.91) and had greater odds of testing positive

**Table 3. Consequences of alcohol use among RCCS participants between March 2017 and September 2018 who reported to be pregnant and used alcohol in the past year, by alcohol use during pregnancy and IPV.**

| | Total | Used alcohol during pregnancy | | | Experienced IPV, past year | | |
|---|---|---|---|---|---|---|---|
| | N = 386 | Yes (n = 290) | No (n = 96) | | Yes (n = 172) | No (n = 214) | |
| | | n (%) | n (%) | p-value | n (%) | n (%) | p-value |
| **No consequences** | 280 (73) | 210 (72) | 70 (73) | 0.518 | 97 (56) | 183 (86) | <0.001 |
| **Alcohol-Related Aggression & Violence** | | | | | | | |
| *In the past year when you drank alcohol, did you ever*: | | | | | | | |
| Get angry | 37 (10) | 30 (10) | 7 (7) | 0.378 | 34 (20) | 3 (1) | <0.001 |
| Get violent/in a fight | 30 (8) | 27 (9) | 3 (3) | 0.052 | 27 (16) | 3 (1) | <0.001 |
| **Harmful Alcohol Use** | | | | | | | |
| *In the past year when you drank alcohol, did you ever*: | | | | | | | |
| Feel ashamed of something done | 25 (6) | 18 (6) | 7 (7) | 0.708 | 12 (12) | 4 (2) | <0.001 |
| Forget things you did/happened | 18 (5) | 15 (5) | 3 (3) | 0.41 | 15 (9) | 3 (1) | 0.001 |
| **Alcohol Dependence** | | | | | | | |
| *In the past year when you drank alcohol, did you ever*: | | | | | | | |
| Have an unsteady gait | 46 (12) | 28 (10) | 18 (19) | 0.017 | 33 (19) | 13 (6) | <0.001 |
| Fall over | 3 (1) | 3 (1) | 0 (0) | 0.317 | 3 (2) | 0 (0) | 0.088 |
| Have difficulty speaking | 12 (3) | 8 (3) | 4 (3) | 0.491 | 12 (7) | 0 (0) | <0.001 |
| Have shaking hands the next morning | 12 (3) | 8 (3) | 4 (4) | 0.491 | 10 (6) | 2 (1) | 0.007 |
| Fail to something wanted/needed | 51 (13) | 40 (14) | 11 (11) | 0.566 | 35 (21) | 16 (7) | <0.001 |
| **Loss of Control Related to Drinking** | | | | | | | |
| In the past year, did you feel you should cut down on your drinking or stop? | 133 (34) | 104 (36) | 29 (30) | 0.312 | 72 (42) | 61 (29) | 0.007 |
| **Alcohol's Impact on Daily Responsibilities and Activities** | | | | | | | |
| In the past year have you taken alcohol while at work? | 49 (13) | 42 (14) | 7 (7) | 0.067 | 28 (16) | 21 (10) | 0.066 |

Note: *p-values were derived from Fisher's Exact tests and reflect the relationship between alcohol consequences and those who did and did not report IPV (emotional, physical, and/or sexual).

for HIV or an STI (AOR: 5.87; 95% CI: 1.99–17.35). Previous research among women in Rakai found that alcohol use before sex was associated with HIV incidence (one partner drank: adjIRR 1.40, 95% CI 1.02–1.92; both partners drank: adjIRR 1.81, 95% CI 1.34–2.45), physical IPV (at least one partner drank: AOR 1.68; 95% CI: 1.41–2.01) [9], and sexual coercion (at least one partner drank: AOR 1.85, 95% CI 1.48–2.31) [18]. A study in South Africa found that mothers with depression, who experienced IPV, and/or who were HIV infected were more likely to drink alcohol [19]. The accumulation of health problems among women in SSA demonstrates a need to understand how multiple conditions interact to increase risks for adverse outcomes.

## Next steps

Our finding of high alcohol use during pregnancy in Rakai, Uganda suggests the need for additional research on maternal alcohol use, neonatal health outcomes, and interventions to address these global public health problems. First, we recommend future research on maternal alcohol use utilize a standardized, validated psychometric tool to measure alcohol consumption quantity and frequency. According to the U.S. Centers for Disease Control and Prevention, there is no known safe amount of alcohol use during pregnancy and no safe time during

pregnancy to drink [20]. However, quantity, frequency, and timing impact risks for adverse health outcomes, including HIV transmission and FASD. A recent systematic review and meta-analysis found that disease and mortality outcomes associated with alcohol use were often accelerated in a dose-response relationship [21]. This study also found that heavy drinking levels and alcohol use disorders were associated with viral load increases, which were partly mediated by treatment non-adherence [21]. Another study found that, compared to no reported drinking during pregnancy, the odds of FASD increased with the number of trimesters that a woman used alcohol [22]. This study also found that binge drinking was associated with increased risk for FASD. Thus, research that includes drinking patterns will be important for developing appropriate prevention strategies.

The burden of FASD in SSA may be large, but it is not well measured. FASD is an umbrella term describing the range of effects that can occur when an individual is antenatally exposed to alcohol, with the most serious being fetal alcohol syndrome [23]. FASD is associated with serious and long-term effects on children [5,23], which can result in significant economic impact [24]. There are currently no data on FASD in Uganda [25]. One modeling study estimated that the prevalence of FASD in Uganda was 16.2 (95% CI: 10–24.3) per 1000 population [25]. Given our finding that a large proportion of pregnant women reported drinking during pregnancy, there is an urgent need to generate awareness and increase training in FASD diagnosis in order to increase surveillance and link children to care.

Maternal alcohol use and corresponding adverse outcomes are preventable. Thus, implementing screening, brief interventions, and referrals for alcohol use and IPV could result in important benefits for women and children in Uganda [17,19]. While some interventions have shown promise for reducing harmful alcohol use among pregnant women [26,27], including Case Management for women at high risk for having a child with FASD; more research is needed demonstrating intervention effectiveness in Uganda [28,29]. Additionally, there is some evidence showing alcohol reduction policies and programs are cost-effective [30,31]; however, among pregnant women, the research is limited. In low resource settings, it may be possible to prioritize interventions for women with high risk for using alcohol during pregnancy. For example, we found that pregnant women who work in bars or restaurants were at increased risk. Further, leveraging antenatal care infrastructure, integrating multiple services to address the syndemics of alcohol use, IPV, and HIV; and including community-based support and task-shifting may increase the value of programs that have the potential to promote healthier pregnancies in SSA.

This study had several limitations. First, our data were cross-sectional and we were unable to assess causality in the relationships between alcohol use during pregnancy and correlates. Next, alcohol use, number of sex partners, and IPV were self-reported and participants might have underestimated or forgotten their true behaviors, potentially leading to reduced accuracy of findings. However, it is encouraging to note that our alcohol use findings are similar to previous studies, some of which used biomarkers [7]. Third, our analysis is among women who self-reported as pregnant and our sample included some women who were not pregnant and excluded women who were unknowingly pregnant. Thus, our findings represent the behavior of women who believed they were pregnant; however, longitudinal studies that include neonatal health outcomes should include participants with confirmed pregnancies. Fourth, the alcohol consequence and alcohol use by sex partners questions reflect alcohol use over the past year and we were unable to determine whether they took place during pregnancy or whether behavior changed after the pregnancy. Finally, although our alcohol use measures were based on items from the internationally validated AUDIT, the questions were not taken verbatim and the full tool was not included in RCCS. Thus, we were unable to fully capture the quantity or frequency of alcohol consumed and related severity or intensity of alcohol use. We

recommend future studies utilize standardized, validated psychometric tools for alcohol use such as the AUDIT, AUDIT-C, TWEAK (Tolerance, Worried, Eye-opener, Amnesia, Cut Down) and the Substance Use Risk Profile-Pregnancy (SURP-P) as well as an alcohol bio-marker to improve the validity of the measurement.

## Conclusion

This study fills an important gap in the literature by estimating the prevalence and correlates of alcohol use during pregnancy using a recent cohort of participants in the RCCS. We found that alcohol use during pregnancy was common and was associated with religion, occupation, higher numbers of past year sex partners, having a partner who drank before sex in the past 12 months, and IPV experience. More research is needed to understand the quantity, frequency, and timing of antenatal alcohol use; and potential impacts on neonates; and to identify programs and services that are acceptable and effective among pregnant women.

## Author Contributions

**Conceptualization:** Adriane Wynn, Dorean Nabukalu, Tom Lutalo, Maria Wawer, David M. Serwadda, Nelson Sewankambo, Fred Nalugoda, Godfrey Kigozi, Jennifer A. Wagman.

**Data curation:** Adriane Wynn, Dorean Nabukalu, Tom Lutalo, Jennifer A. Wagman.

**Formal analysis:** Adriane Wynn, Dorean Nabukalu, Tom Lutalo, Larry W. Chang, Jennifer A. Wagman.

**Funding acquisition:** Adriane Wynn, Jennifer A. Wagman.

**Investigation:** Jennifer A. Wagman.

**Methodology:** Adriane Wynn, Jennifer A. Wagman.

**Project administration:** David M. Serwadda, Nelson Sewankambo, Fred Nalugoda, Jennifer A. Wagman.

**Supervision:** Maria Wawer, Larry W. Chang, Susan M. Kiene, David M. Serwadda, Nelson Sewankambo, Fred Nalugoda, Godfrey Kigozi.

**Validation:** Maria Wawer, Larry W. Chang, Susan M. Kiene, David M. Serwadda, Nelson Sewankambo, Fred Nalugoda, Godfrey Kigozi.

**Writing – original draft:** Adriane Wynn, Jennifer A. Wagman.

**Writing – review & editing:** Adriane Wynn, Dorean Nabukalu, Tom Lutalo, Maria Wawer, Larry W. Chang, Susan M. Kiene, David M. Serwadda, Nelson Sewankambo, Fred Nalugoda, Godfrey Kigozi, Jennifer A. Wagman.

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
