## [Decision Letter · Decision Letter 0]

5 Feb 2021

PONE-D-21-02267

Alcohol use during pregnancy in Rakai, Uganda.

PLOS ONE

Dear Dr. Wynn,

Thank you for submitting your manuscript to PLOS ONE. After careful consideration, we feel that it has merit but does not fully meet PLOS ONE’s publication criteria as it currently stands. Therefore, we invite you to submit a revised version of the manuscript that addresses the points raised during the review process.

We look forward to receiving your revised manuscript.

Kind regards,

Wendee Wechsberg

Academic Editor

PLOS ONE

Journal Requirements:

2. Please include your tables as part of your main manuscript and remove the individual files. Please note that supplementary tables (should remain/ be uploaded) as separate "supporting information" files

3. Please consider including your recent study (Wagman, Jennifer A., et al. "Prevalence and correlates of men’s and women’s alcohol use in agrarian, trading and fishing communities in Rakai, Uganda." Plos one 15.10 (2020): e0240796.3) in the discussion and/or Introduction sections.

Reviewers' comments:

Reviewer's Responses to Questions

**Comments to the Author**

1. Is the manuscript technically sound, and do the data support the conclusions?

Reviewer #1: Yes

Reviewer #2: Yes

Reviewer #3: Yes

2. Has the statistical analysis been performed appropriately and rigorously? 

Reviewer #1: I Don't Know

Reviewer #2: Yes

Reviewer #3: Yes

3. Have the authors made all data underlying the findings in their manuscript fully available?

Reviewer #1: Yes

Reviewer #2: No

Reviewer #3: Yes

4. Is the manuscript presented in an intelligible fashion and written in standard English?

Reviewer #1: Yes

Reviewer #2: Yes

Reviewer #3: Yes

5. Review Comments to the Author

Reviewer #1: The study contributes to the body of literature on global prenatal alcohol use and syndemic issues by addressing the prevalence and associated factors in Uganda populations. The manuscript needs minor editing and modifications.

Introduction

-There is nothing stated about HIV prevalence and the status quo in Uganda besides alcohol use, while the abstract mentions both issues.

-Please state where is Kampala, is this a country or city?

Method

-Please state the rationale of why asking prenatal alcohol use questions only among those who self reported pregnancy and excluded those who tested positive of pregnancy but were unaware of pregnancy.

-Please add to future research agenda to examine the difference between women who were aware of pregnancy but consumed alcohol versus those who were not aware of pregnancy and consumed alcohol during the current pregnancy.

-Please clearly state that 21 who tested negative of pregnancy but self reported pregnancy were also excluded.

-I do not know if authors need to state about comparing differences between those who tested negative versus positive of pregnancy because the study did not include those who tested negative of pregnancy.

-There is a quotation mark “ that is not necessary at the end of the first paragraph of Method.

Results/Discussion

-Table 1 can bold p values that showed significant differences so readers do not have to search.

-It is very interesting that smoking was not prevalent among those who reported alcohol use during the current pregnancy, because it contradicts with the literature at least in USA and South Africa. Please address this important finding in a paragraph of Discussion.

-Future research on maternal alcohol use with a validated psychometric tool should bring up screening tools specifically made for prenatal alcohol use risk such as TWEAK and Substance Use Risk Profile-Pregnancy (SURP-P).

-There are more intensive community-based behavioral interventions to reduce alcohol use during pregnancy done in South Africa that showed effects in reduced drinking levels, which need to be introduced in Discussion (de Vries et al., 2015; May et al., 2013).

de Vries, M. M., Joubert, B., Cloete, M., Roux, S., Baca, B. A., Hasken, J. M., … May, P. A. (2015). Indicated Prevention of Fetal Alcohol Spectrum Disorders in South Africa: Effectiveness of Case Management. International Journal of Environmental Research and Public Health, 13(1), ijerph13010076. https://doi.org/10.3390/ijerph13010076

May, P. A., Marais, A.-S., Gossage, J. P., Barnard, R., Joubert, B., Cloete, M., … Blankenship, J. (2013). Case Management Reduces Drinking During Pregnancy among High Risk Women. The International Journal of Alcohol and Drug Research, 2(3), 61–70. https://doi.org/10.7895/ijadr.v2i3.79

Reviewer #2: Thank you for the opportunity to review this manuscript.

This paper provides a valuable contribution to the literature. It is relevant globally, but particularly across the African continent where similar drinking patterns in the population exists and in pregnant women specifically and where synergistic impact of alcohol, IPV and HIV play a big role.

Minor recommendations/revision:

there are parts of the methods that are unclear.

The description of the RCCS are slightly confusing. Are individuals surveyed once off? Are there repeat visits ("at each survey visit" is mentioned). Is the census for the sole purpose of identification of participants or part of a larger population census? What is considered for eligibility?

Were the 21 women who tested negatively for HCG ultimately excluded. I don't think so, but clarify this in the text where it is mentioned.

Please provide some context as to what the resident community types mean. Explain that Ugandan communities are split into the three types and why etc.

Reviewer #3: This is an important, well written manuscript about alcohol use during pregnancy in Rakai, Uganda. There are several areas where the paper and analyses could be strengthened prior to publication.

1. The data collection around alcohol- were those questions validated or previously used in this setting? Why were AUDIT questions not used? Was quantity of alcohol consumed ascertained? If so please present this and if not please list as an important limitation of this analysis.

2. Abstract- please include age and gestational age of women in the study. Is IPV - ever IPV or in past 12m? What were reference categories for catholic religion (why is it other religion, small group?), and other refs? What did models adjust for? Finally, your data don't show an association between HIV and alcohol use (Results: "Marital status, education level, past year smoking, and HIV status were no longer associated with

alcohol use during pregnancy after adjusting for covariates.")-- please update the conclusion.

3. Introduction- para 2, reference 6, can you include the quantity of alcohol consumed in the study if available (similar comment for other referenced studies)?

4. Methods- similar to question above, were questions used previously validated in this population? If not, how were they tested/translated? What were the authors' hypotheses about covariates associated with alcohol use? Ideally the analysis would be hypothesis drive and each model with adjust for covariates that may be confounders in the models, instead of just putting in various variables that have p<0.05 and may not be associated with alcohol and the covariate (e.g. may introduce bias in the model).

5. Results - see concerns above about quantity of alcohol consumed and multivariate models and hypotheses

Table 3- did the authors run multivariate models for the alcohol and IPV models here or only univariate models? If so, please present data on mulitvariate analyses.

6. Discussion- Para 1 about global average of drinkers, is this in pregnant women or all avg? What about the uganda popn average? Would be good to compare with local data as well. Please include frequency and quantity in para 3 if available. Similarly, the study did not quantify drinking behaviors (frequency, location, quantity) so please update the discussion around this in final para in discussion.

6. PLOS authors have the option to publish the peer review history of their article (what does this mean?). If published, this will include your full peer review and any attached files.

Reviewer #1: No

Reviewer #2: No

Reviewer #3: No

---

## [Author Response · Author response to Decision Letter 0]

18 Mar 2021

Reviewer #1: The study contributes to the body of literature on global prenatal alcohol use and syndemic issues by addressing the prevalence and associated factors in Uganda populations. The manuscript needs minor editing and modifications.

Introduction

1. There is nothing stated about HIV prevalence and the status quo in Uganda besides alcohol use, while the abstract mentions both issues.

We have removed the mention of HIV in the introduction section of the abstract. 

2. Please state where is Kampala, is this a country or city?

We have clarified that Kamala is the capital of Uganda.

Method

1. Please state the rationale of why asking prenatal alcohol use questions only among those who self reported pregnancy and excluded those who tested positive of pregnancy but were unaware of pregnancy.

In the main RCCS study, the questions related to alcohol use during pregnancy were only asked among women who self-reported as pregnant. Thus, we do not have data on alcohol use during pregnancy among women who were pregnant, but unaware. Our results reflect the alcohol use behaviors of women who believed they were pregnant. 

2. Please add to future research agenda to examine the difference between women who were aware of pregnancy but consumed alcohol versus those who were not aware of pregnancy and consumed alcohol during the current pregnancy.

We have bolstered our limitations and next steps sections to call for more studies on alcohol use among women who are pregnant, but unaware, which is important for research related to neonatal health outcomes.

3. Please clearly state that 21 who tested negative of pregnancy but self reported pregnancy were also excluded.

We have clarified that the 21 women who tested negative for pregnancy were included in our sample. We have clarified that our sample is women who self-report as pregnant and our results reflect alcohol use among women who believe they are pregnant.

4. I do not know if authors need to state about comparing differences between those who tested negative versus positive of pregnancy because the study did not include those who tested negative of pregnancy.

We compared those who tested negative with those who tested positive because all women who self-reported as pregnant were included in our sample. 

5. There is a quotation mark “ that is not necessary at the end of the first paragraph of Method.

Thank you, we have removed the quotation mark.

Results/Discussion

1. Table 1 can bold p values that showed significant differences so readers do not have to search.

Done.

2. It is very interesting that smoking was not prevalent among those who reported alcohol use during the current pregnancy, because it contradicts with the literature at least in USA and South Africa. Please address this important finding in a paragraph of Discussion.

Smoking was not prevalent among our entire sample of women who believed they were pregnant (18 women reported smoking out of 960). However, smoking was associated with increased odds for alcohol use during pregnancy. Previous research in Rakai, Wagman, Jennifer A., et al. "Prevalence and correlates of men’s and women’s alcohol use in agrarian, trading and fishing communities in Rakai, Uganda." (2020,) also found tobacco smoking was rare among women (2.7% of women in their sample of 10,010 reported smoking). 

3. Future research on maternal alcohol use with a validated psychometric tool should bring up screening tools specifically made for prenatal alcohol use risk such as TWEAK and Substance Use Risk Profile-Pregnancy (SURP-P).

Thank you. In the limitations section, we have added the suggestion that future research utilize validated psychometric tools such as the AUDIT, AUDIT-C, TWEAK (Tolerance, Worried, Eye-opener, Amnesia, Cut Down) and the Substance Use Risk Profile-Pregnancy (SURP-P) as well as an alcohol biomarker to improve the validity of the measurement

4. There are more intensive community-based behavioral interventions to reduce alcohol use during pregnancy done in South Africa that showed effects in reduced drinking levels, which need to be introduced in Discussion (de Vries et al., 2015; May et al., 2013).

Thank you for providing these studies on Case Management to reduce alcohol use during pregnancy and FASD. We have cited these studies in our discussion and clarified that interventions are needed in Uganda.

Reviewer #2: Thank you for the opportunity to review this manuscript. This paper provides a valuable contribution to the literature. It is relevant globally, but particularly across the African continent where similar drinking patterns in the population exists and in pregnant women specifically and where synergistic impact of alcohol, IPV and HIV play a big role.

Minor recommendations/revision:

1. The description of the RCCS are slightly confusing. Are individuals surveyed once off? Are there repeat visits ("at each survey visit" is mentioned). Is the census for the sole purpose of identification of participants or part of a larger population census? What is considered for eligibility?

The RCCS is an open cohort study and communities have been continuously evaluated since 1994. At each time point, a household census is conducted to identify eligible individuals. Thus, it is likely that individuals will be interviewed repeatedly as part of the study, but not assured. Individuals are eligible if they are between 15 to 49 years and residents of the study communities.

We have included more detail about the RCCS in the methods section.

2. Were the 21 women who tested negatively for HCG ultimately excluded. I don't think so, but clarify this in the text where it is mentioned.

We have clarified that our sample includes all women who self-reported as pregnant and thus includes the 21 who later tested negative.

3. Please provide some context as to what the resident community types mean. Explain that Ugandan communities are split into the three types and why etc.

We have included more detail about the resident community types in the methods/measures section.

Reviewer #3: This is an important, well written manuscript about alcohol use during pregnancy in Rakai, Uganda. There are several areas where the paper and analyses could be strengthened prior to publication.

1. The data collection around alcohol- were those questions validated or previously used in this setting? Why were AUDIT questions not used? Was quantity of alcohol consumed ascertained? If so please present this and if not please list as an important limitation of this analysis.

Although our alcohol use measures were based on items from the internationally validated AUDIT, the questions were not taken verbatim and the full tool was not included in RCCS. Thus, we were unable to fully capture the quantity or frequency of alcohol use. As you suggest, we have included this as a limitation in our discussion section.

2. Abstract- please include age and gestational age of women in the study. Is IPV - ever IPV or in past 12m? What were reference categories for catholic religion (why is it other religion, small group?), and other refs? What did models adjust for? Finally, your data don't show an association between HIV and alcohol use (Results: "Marital status, education level, past year smoking, and HIV status were no longer associated with alcohol use during pregnancy after adjusting for covariates.")-- please update the conclusion.

We included the median age in the results, however, gestational age was not collected. We added that IPV was measured over the past 12 months. We added the model covariates and the variable reference categories to the abstract results. We also updated the conclusion.

Introduction

1. para 2, reference 6, can you include the quantity of alcohol consumed in the study if available (similar comment for other referenced studies)?

Unfortunately, we could not ascertain the quantity of alcohol consumed and have included this in our limitations section.

Methods

1. similar to question above, were questions used previously validated in this population? If not, how were they tested/translated? What were the authors' hypotheses about covariates associated with alcohol use? Ideally the analysis would be hypothesis drive and each model with adjust for covariates that may be confounders in the models, instead of just putting in various variables that have p<0.05 and may not be associated with alcohol and the covariate (e.g. may introduce bias in the model).

We included covariates in the model based on our hypotheses derived from previous research in Uganda, which found that alcohol use was associated with HIV, IPV, occupation, smoking status, and sex partner characteristics. 

2. Results - see concerns above about quantity of alcohol consumed and multivariate models and hypotheses

Please see our responses above.

3. Table 3- did the authors run multivariate models for the alcohol and IPV models here or only univariate models? If so, please present data on mulitvariate analyses.

In Table 3, we only assessed bivariate comparisons and have clarified this in the Methods/analysis section. 

Discussion

1. Para 1 about global average of drinkers, is this in pregnant women or all avg? What about the uganda popn average? Would be good to compare with local data as well. Please include frequency and quantity in para 3 if available. Similarly, the study did not quantify drinking behaviors (frequency, location, quantity) so please update the discussion around this in final para in discussion.

We clarified that the global average is among pregnant women and that our finding of 35% was higher than the previous studies among pregnant women in other regions of Uganda (discussed in the introduction). However, our finding was lower than the prevalence found in a previous study among non-pregnant women in Rakai. We were not able to assess quantity/frequency and have discussed these limitations in the discussion.

---

## [Decision Letter · Decision Letter 1]

9 Aug 2021

Alcohol use during pregnancy in Rakai, Uganda.

PONE-D-21-02267R1

Dear Dr. Wynn,

We’re pleased to inform you that your manuscript has been judged scientifically suitable for publication and will be formally accepted for publication once it meets all outstanding technical requirements.

Kind regards,

Janet E Rosenbaum, Ph.D.

Academic Editor

PLOS ONE

Additional Editor Comments (optional):

Reviewers' comments:

Reviewer's Responses to Questions

**Comments to the Author**

1. If the authors have adequately addressed your comments raised in a previous round of review and you feel that this manuscript is now acceptable for publication, you may indicate that here to bypass the “Comments to the Author” section, enter your conflict of interest statement in the “Confidential to Editor” section, and submit your "Accept" recommendation.

Reviewer #1: All comments have been addressed

Reviewer #2: All comments have been addressed

Reviewer #3: All comments have been addressed

2. Is the manuscript technically sound, and do the data support the conclusions?

Reviewer #1: Yes

Reviewer #2: Yes

Reviewer #3: Yes

3. Has the statistical analysis been performed appropriately and rigorously? 

Reviewer #1: I Don't Know

Reviewer #2: Yes

Reviewer #3: Yes

4. Have the authors made all data underlying the findings in their manuscript fully available?

Reviewer #1: Yes

Reviewer #2: Yes

Reviewer #3: Yes

5. Is the manuscript presented in an intelligible fashion and written in standard English?

Reviewer #1: Yes

Reviewer #2: Yes

Reviewer #3: Yes

6. Review Comments to the Author

Reviewer #1: all comments are adequately addressed, and the study is ready for a publication now. There are no further comments.

Reviewer #2: (No Response)

Reviewer #3: No further comments for the author. The revisions have strengthened the paper and is good to go after another round of edits for consistency.

7. PLOS authors have the option to publish the peer review history of their article (what does this mean?). If published, this will include your full peer review and any attached files.

Reviewer #1: No

Reviewer #2: No

Reviewer #3: No

---

## [Editor Report · Acceptance letter]

17 Aug 2021

PONE-D-21-02267R1 

Alcohol use during pregnancy in Rakai, Uganda. 

Dear Dr. Wynn:

I'm pleased to inform you that your manuscript has been deemed suitable for publication in PLOS ONE. Congratulations! Your manuscript is now with our production department. 

Kind regards, 

on behalf of

Dr. Janet E Rosenbaum 

Academic Editor

PLOS ONE